# Differentiation of the bacterial communities associated with *Orbicella faveolata* across different growth conditions and life-cycle stages

Amanda Pérez-Trejo[1], Ma. Leopoldina Aguirre-Macedo[1], Anastazia T. Banaszak[2]*, José Q. García-Maldonado[1]*

1 Departamento Recursos del Mar, Centro de Investigación y de Estudios Avanzados del Instituto Politécnico Nacional, Unidad Mérida, Mérida, Yucatán, México, 2 Unidad Académica Puerto Morelos, Instituto de Ciencias del Mar y Limnología, Universidad Nacional Autónoma de México, Puerto Morelos, Mexico

* banaszak@cmarl.unam.mx (ATB); jose.garcia@cinvestav.mx (JQGM)

## Abstract

The coral microbiome can strongly influence coral health, development, and resilience. While larval settlement is fundamental for coral restoration efforts using assisted larval propagation, post-settlement survival remains a major challenge. The study of lab-bred *Orbicella faveolata* settlers (LBOFS) microbiome has been proposed due to its potential role in coral adaptation processes. However, there is limited information about LBOFS bacterial communities and comparisons between different growth conditions and life-cycle stages have not been conducted. Using 16S rRNA high-throughput sequencing, we analyzed the structure and composition of LBOFS-associated bacteria and compared them to those from outplanted LBOFS and wild settlers. We also compared the microbiomes of settlers to adult colonies. The LBOFS bacterial community was composed of 4224 ASVs with the Orders Kiloniellales, Rhodobacterales, Cytophagales, Cyanobacteriales, and Flavobacteriales being the most abundant across the samples, with a rare biosphere consisting of 44.6% relative abundance. A Principal Coordinates Analysis and a PERMANOVA indicated significantly different bacterial community structures based on settler growth conditions and life-cycle stage. Linear discriminant analysis Effect Size analysis identified specific taxa whose differential abundances contributed to the observed differences. For settler growth conditions, the differences were mainly due to the Order Cyanobacteriales for LBOFS, SAR202 clade for outplanted settlers, and Microtrichales for wild samples. Statistical analysis of functional prediction showed significant differences only in nitrogen fixation for LBOFS. For life-cycle stage, LEfSe revealed that the Orders Cytophagales and Cyanobacteriales exhibited the highest differential abundances in adults and settlers, respectively. Functional prediction revealed that nitrogen fixation and oxygenic photoautotrophy were more enriched in settlers, whereas nitrate reduction and anaerobic chemoheterotrophy were more

**Data availability statement:** The raw sequencing data generated in this study have been deposited at NCBI under the PRJNA1244432 BioProject accession number.

**Funding:** This research was supported by Consejo Nacional de Ciencia y Tecnología (CONACYT) through grant FORDECYT-PRONACES, CF-2019-425888 to A.T.B. and J.Q.G.-M. We thank Secretaria de Ciencia, Humanidades, Tecnología e Innovación (Secihti) for providing the doctoral scholarship to APT (627723) during the development of this study. The funding institution, Secihti, had no role in study design, data collection and analysis, decision to publish, or manuscript preparation.

**Competing interests:** The authors have declared that no competing interests exist.

enriched in adults. This study highlighted the bacterial taxa and predicted metabolic processes that could potentially contribute to coral settler functioning, providing a valuable baseline for future research to enhance their survival rates using probiotics.

## Introduction

Coral-microbiome associations across all life stages are essential for survival, particularly in a changing environment. Larvae settlement is fundamental to coral restoration efforts using assisted larval propagation techniques. However, larval post-settlement survival is a major bottleneck due to the susceptibility of settlers to predation, competition, and stochastic disturbances. Improving larval growth rates could help them pass through this vulnerable phase [1,2]. The analysis of the microbiome could be crucial for enhancing coral function and used to improve efforts to rescue threatened species [3]. In addition, the response of the microbiome to stressful conditions and environmental changes enables the identification of key host-microbiome associations, which may be essential for coral resilience and adaptation [4–6]. Coral microbiome monitoring and manipulation can be used to effectively manage and maintain organismal health [3].

Bacteria are the dominant group in the coral microbiome as they contribute to stress tolerance and adaptation to a changing environment [7]. This occurs through mechanisms including the promotion of coral nutrition and growth, mitigation of stress and toxic compound impacts, deterrence of pathogens, and benefiting early life-stage development [8]. During the early-life stages, beneficial traits include the production of signals for the modulation or regulation of larval settlement, the production of biofilm, which facilitates larval settlement, and quorum quenching or the breakdown of quorum-sensing molecules, which disrupt cell-cell signaling of beneficial microbes for other species competing for space [8].

As previously reviewed by [9] practices associated with coral breeding for reef restoration focus on improving settler health prior to outplanting on the reef, including the use of beneficial microbes. This approach requires studying the microbiome during the early life stages of corals to potentially improve larval post-settlement survival and growth using probiotics.

Previous studies on *O. faveolata* have shown that microbial communities associated with crustose coralline algae (CCA) enhance larval settlement [10,11]. In reef environments, degrading conditions can reduce CCA cover and may compromise this process. Other studies conducted in different coral species at early life stages reported the promotion of larval settlement and enhancement of post-settlement survival using probiotics. For example, when administering the strain cB07 of *Metabacillus indicus* to *Pocillopora damicornis*. Higher post-settlement survival was exhibited by the coral *Porites astreoides* when settled onto bacteria biofilm as well as on coralline algae [12]. In another study, larval settlement/metamorphosis of *Pocillopora damicornis* was strongly correlated with the bacterial community composition of diverse calcified algae [13]. Bacterial strains isolated from the crustose coralline algae *Hydrolithon reinboldii* were tested on *Leptastrea purpurea* larvae and found to enhance settlement behavior [14].

Based on the above, there is limited information on bacterial communities associated with settlers bred under culture conditions and their ability to improve settler health once outplanted onto coral reefs. Therefore, as a first step, this study aimed to: (1) identify and analyze the microbial communities associated with *O. faveolata* lab-bred settlers (LBOFS), (2) compare the bacterial communities associated with LBOFS grown under different conditions, and (3) compare life-cycle stages (settlers vs wild adults). With this information, we aimed to identify key bacterial taxa and metabolic processes to be considered for further research and testing of microbial consortia to improve post-settlement survival rates and growth once outplanted onto the reef.

## Methods

### Sample collection and breeding conditions

During the spawning of *O. faveolata*, gametes were collected and carefully transported to the CORALIUM Laboratory at the Unidad Academica de Sistemas Arrecifales in Puerto Morelos, Quintana Roo, Mexico (Collection permit issued to Universidad Nacional Autónoma de México (PPF/DGOPA-070/20). Upon arrival, assisted fertilization was applied to promote the fusion of gametes from distinct colonies, to generate new individuals.

Embryonic development was monitored by microscopy post-fertilization. Embryos were transferred to incubators and maintained in the laboratory until larval development, which occurred 20–32 hours after fertilization. Incubators were maintained at 28°C with a 12:12 hours light:dark photoperiod.

Larval settlement occurred between 3 and 7 days post-fertilization. During this stage conditioned substrates were introduced into the incubators to promote settlement. These substrates were composed of a mixture 50:50 of sea sand and concrete, that were previously conditioned on the reef for 60 days, to allow for the attachment of microorganisms relevant to larval settlement.

After settlement, they were transferred to outdoor aquarium facilities where light intensity was kept below 400 µmol m$^{-2}$ s$^{-1}$ during the first two months, with gradual increases thereafter. Salinity was maintained at 35–36 PSU throughout this period. All this information can be found in [15].

### Sample processing and DNA extraction

From a total of 34 samples of *O. faveolata,* 16 samples corresponded to lab-bred settlers (LBOFS) (PCF01-PCF16). The remaining samples were collected from Jardines Reef (20° 49' 51.53" N 86° 52' 29.92" W) at a depth of 9 m in the Puerto Morelos Reef National Park, Mexico in 2022. For adult colonies, 12 fragments (< 1 cm$^2$) were collected (PCF21-PCF32). For outplanted settlers, 3 samples from the 2019 cohort were collected (PCF734-PCF736), along with 3 additional samples from wild settlers (< 1 cm$^2$) (PCF737-PCF739). The settlers were carefully detached from the substrate using a sterile scalpel and placed into Eppendorf tubes with RNAlater solution (Qiagen, Hilden, Germany) to preserve genetic material, as were the samples of the adult colonies.

DNA extraction was performed utilizing the Blood and Tissue Kit (Qiagen, Hilden, Germany) following the manufacturer's instructions with some modifications. Specifically, the dry bath incubator (Major Science Co., Taoyuan, Taiwan) was used for 30 minutes due to the fragment size or until lysis of symbionts was observed, resulting in a yellow-ochre coloration. The DNA was then fixed in 50 µl of DNA buffer and used for 16S rRNA gene amplification.

### 16s rRNA library preparation and sequencing

Library preparation was performed following the *16S Metagenomic Sequencing Library Preparation from Illumina MiSeq System* protocol, utilizing two primer sets designed to target the bacterial 16S rRNA gene V3 and V4 regions as selected from [16] consisting of the forward primer S-D-Bact-0341-b-S-17 5′-CCTACGGGNGGCWGCAG-3′ and the reverse primer S-D-Bact-0785-a-A-21, 5′-GACTACHVGGGTATCTAATCC-3′. The polymerase chain reaction (PCR) consisted of a 20 µl

reaction using 10 µl of Phusion Flash High-Fidelity Master Mix (Thermo Scientific, Waltham, MA, USA), 0.5 µl Forward Primer (10µM), 0.5 µl Reverse Primer (10µM), 2 µl of DNA template and 7 µl nuclease-free water. PCR reaction cycles were performed by denaturation for 3 min at 95°C, 28 cycles at 95°C for 30 s, 55°C for 30 s, and 72°C for 30 s, followed by a 5-min extension at 72°C. All products were electrophoresed on a 1% agarose gel for screening.

All PCR products were purified utilizing AMpure XP beads (Beckman Coulter, Inc., Brea, CA, USA) to remove primer dimers between purifications. PCR products were indexed using the Nextera XT Index Kit (Illumina, San Diego, CA, USA). This process consisted of a 25 µl reaction using 12.5 µl Phusion Flash High-Fidelity Master Mix (Thermo Scientific, Waltham, MA, USA), 2 µl Nextera XT Index Primer 1, 2 µl Nextera XT Index Primer 2, 5 µl DNA, and 3.5 µl nuclease- free water. The reactions were performed by denaturation cycles of 3 min at 95°C, 8 cycles of 95°C for 30 s, 55°C for 30 s, and 72°C for 30 s, followed by a 5-min extension at 72°C. The purified amplicons were pooled and quantified using the Qubit 3.0 Fluorometer (Thermo Fisher Scientific, Waltham, MA, USA) with the High Sensitivity DNA assay kit and loaded on Illumina MiSeq sequencer in a 2 × 250 bp paired-end run.

### Bioinformatic analysis

Demultiplexed paired-end 16S rRNA gene sequences were analyzed using the Quantitative Insights into Microbial Ecology pipeline (QIIME2 version QIIME 2 2024.10.1) [17]. The DADA2 (V 1.16) tool was employed to filter and assess the sequencing reading quality with trimming parameters 30–200 [18], while the SILVA 138.1 database was used for taxonomy assignment, resulting in Amplicon Sequence Variant (ASV) [19]. Sequences assigned to the domain Archaea were removed from the analysis.

Within the R environment, the Phyloseq package filter out chloroplast and mitochondrial sequences [20]. For plotting and statistical analysis of alpha and beta diversity, packages such as Ggplot2 and Microbiota Process were utilized [21,22]. The dataset was rarefied to a depth of 4800 prior to the calculation of the abundance tables, Shannon index, and richness values. As part of the bacterial composition assessment the rare biosphere (defined as taxa with relative abundances < 0.1%) was calculated through Microbiota Process analysis. Principal Coordinates Analysis (PCoA) based on Bray-Curtis dissimilarity along with permutational multivariate analysis of variance (PERMANOVA) performed with 999 permutations were used to assess differences in the bacterial community composition.

Through the MicroEco package [23], a differential abundance analysis was carried out by a Linear discriminant analysis Effect Size (LEfSe) [24]. Moreover, functional prediction was conducted by the FAPROTAX database [25] contained within the same package. Additionally, a redundancy analysis (RDA) based on Bray-Curtis dissimilarity, followed by a Kruskal-Wallis test, was applied to the FAPROTAX results to identify significantly different predicted metabolic processes in the bacterial communities.

## Results

In this study, we described LBOFS-associated bacteria by their structure and composition. Furthermore, utilizing LBOFS information, a comparative analysis was conducted by settler growth conditions (outplanted and wild) and by coral life-cycle stages (settlers versus adults). A total of 29 samples were sequenced for the 16S rRNA gene, corresponding 14 to LBOFS, 3 outplanted settlers, 3 wild settlers, and 9 wild adults. The remaining samples were not included due to low read counts. The limited number of wild and outplanted settlers was due to the difficulty in locating them in their natural habitat.

### Characterization of LBOFS associated bacteria

A total of 236,473 reads and 4224 amplicon sequence variants (ASVs) were obtained from 16S rRNA sequencing after quality filtering. Rarefaction curves were obtained (Supplementary Fig 1 in S1 File) and subsequently a total of 40 phyla, 80 classes, 185 orders, 263 families, and 353 genera were identified. The structure of this bacterial community was estimated by Shannon's alpha diversity index (Supplementary Fig 2A in S1 File) exhibiting 5.477 ± 0.24 (mean ± standard

deviation). Proteobacteria, Bacteroidota, and Cyanobacteria were the phyla showing major relative abundances of 52.6%, 14.8% and 9.2% respectively. At the Order level (Fig 1) Kiloniellales (8.9%), Rhodobacterales (7.4%), Cytophagales (7%), Cyanobacteriales (6.2%), and Flavobacteriales (5.6%) were the major groups. The rare biosphere was composed of 44.6% of relative abundance. Taxonomic resolution across samples was possible at the genus level showing the 3 major genera *Woeseia* sp. (5.5%), *Tistlia* sp. (4.3%), and *Muricauda* sp. (2.8%).

The most predicted functions according to FAPROTAX were by energy source, aerobic chemoheterotrophy (15.85%), and anaerobic chemoheterotrophy (0.24%). Carbon cycle processes were also displayed by fermentation, cellulolysis, and hydrocarbon degradation (2.57%, 0.40%, and 0.12% respectively). Nitrogen cycle processes such as nitrogen fixation (1.04%) and nitrate reduction (0.67%) were also displayed. Other important processes for coral development such as oxygenic photoautotrophy (3.76%) were also exhibited (Supplementary Fig 3 in S1 File).

## Comparison of associated bacteria by settler growth condition

According to the Shannon index and ANOVA analysis, Alpha-diversity did not show differences across growth conditions (Supplementary Fig 2A in S1 File). Beta-diversity analysis showed the highest variation was displayed in axis 1 (16%) and axis 2 (6.1%), demonstrating a separation of LBOFS from outplanted versus wild settlers (Fig 2). According to PERMANOVA (Supplementary Table 1A in S1 File), these differences in bacterial community structures are significant ($p \leq 0.001$).

According to the LEfSe analysis the differences in the bacterial structure were primarily due to the relative abundances of Orders Cyanobacteriales (LDA 4.42) in LBOFS, Microtrichales (LDA 4.41) in the wild, and SAR202_clade (LDA 4.07) in outplanted settlers (Fig 3A).

RDA analysis identified bacterial orders whose relative abundances may be associated with the previously predicted metabolic processes. The results showed Axis 1 and 2 explained 42.8% and 20.6% of the variation across the samples,

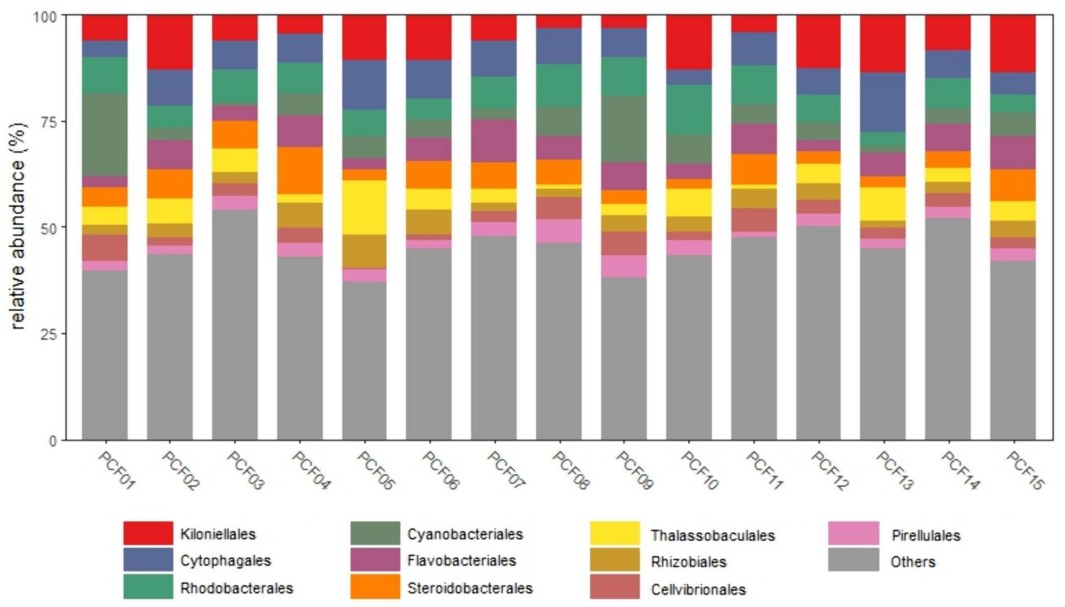

**Fig 1. Bacterial community composition associated with *O. faveolata* lab-bred settler samples (LBOFS).** Rarefied relative abundance (%) of 10 major bacterial Orders with the rare biosphere grouped under "Others".

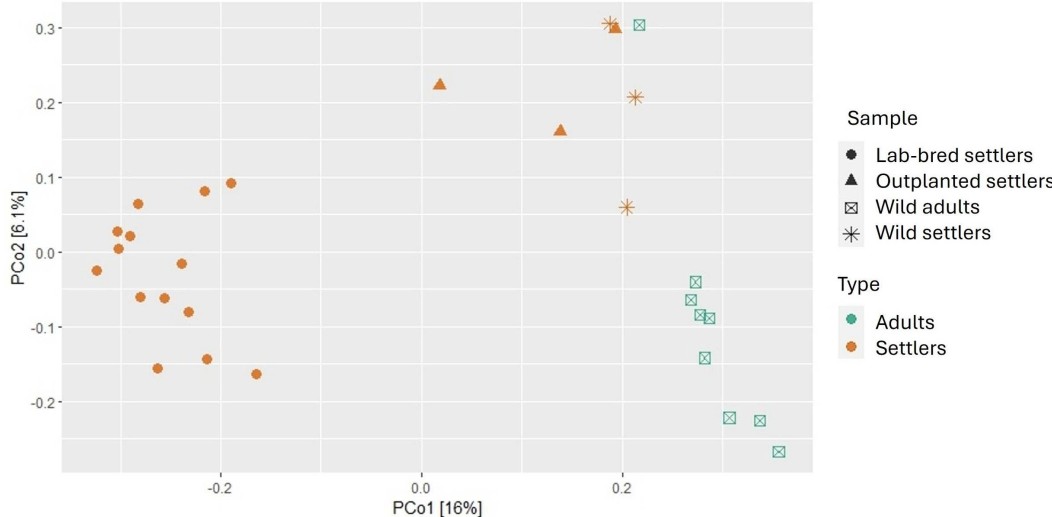

**Fig 2. Beta diversity analysis by PCoA based on Bray-Curtis distances for all *O. faveolata* samples.** In the figure, shape differentiation represents settler growth conditions (PERMANOVA F = 2.37, p = 0.001), while color differentiation corresponds to life-cycle stages (PERMANOVA F = 5.08, p = 0.001), all the samples were rarefied at 4800.

respectively (Fig 4A). According to the Kruskal-Wallis test (Supplementary Table 3 in S1 File), only the nitrogen fixation process was enriched in LBOFS samples ($p = 0.038$).

### Comparison of associated bacteria by life-cycle stage

Alpha-diversity estimated according to the Shannon index exhibited values of 5.53 ± 0.5 for settlers and 5.09 ± 0.5 for adults. The ANOVA showed significant differences between life-cycle stages as shown in (Supplementary Fig 2B in S1 File). According to PCoA based on the Bray-Curtis distances (Fig 2), beta-diversity demonstrated a separation between settlers and adults. According to PERMANOVA (Supplementary Table 1B in S1 File), the differences are significant ($p \leq 0.001$).

Bacterial composition analysis carried out by LEfSe (Fig 3B) showed differences mainly due to the orders of Cytophagales (LDA = 4.5) for adults, and Cyanobacteriales (LDA = 4.2) for settlers.

Functional prediction according to the FAPROTAX database exhibited for adults, aerobic chemoheterotrophy (19.05%), fermentation (3.97%), and nitrate reduction (2.23%) as the most predicted processes, in contrast to nitrogen fixation (0.07%) as the least predicted. For settlers, aerobic chemoheterotrophy (16.41%), fermentation (2.99%), and oxygenic photoautotrophy (2.82%) were the most predicted, contrary to cellulolysis (0.49%) as the least predicted (Supplementary Table 4 in S1 File).

The RDA analysis (Fig 4B), showed that the percentage of explained variation across the samples was 31.7% in Axis 1 and 23.1% in Axis 2. According to the Kruskal-Wallis test, nitrogen fixation ($p = 0.001$) and oxygenic photoautotrophy ($p = 0.002$) were more enriched in settler samples compared to adults. However, nitrate reduction ($p = 0.002$) and anaerobic chemoheterotrophy ($p = 0.002$) were significantly more enriched in adults compared to settler samples (Supplementary Table 5 in S1 File).

### Discussion

This study shows that the bacterial communities associated with the reef-building coral *O. faveolata* differ depending on growth conditions and life-cycle stage.

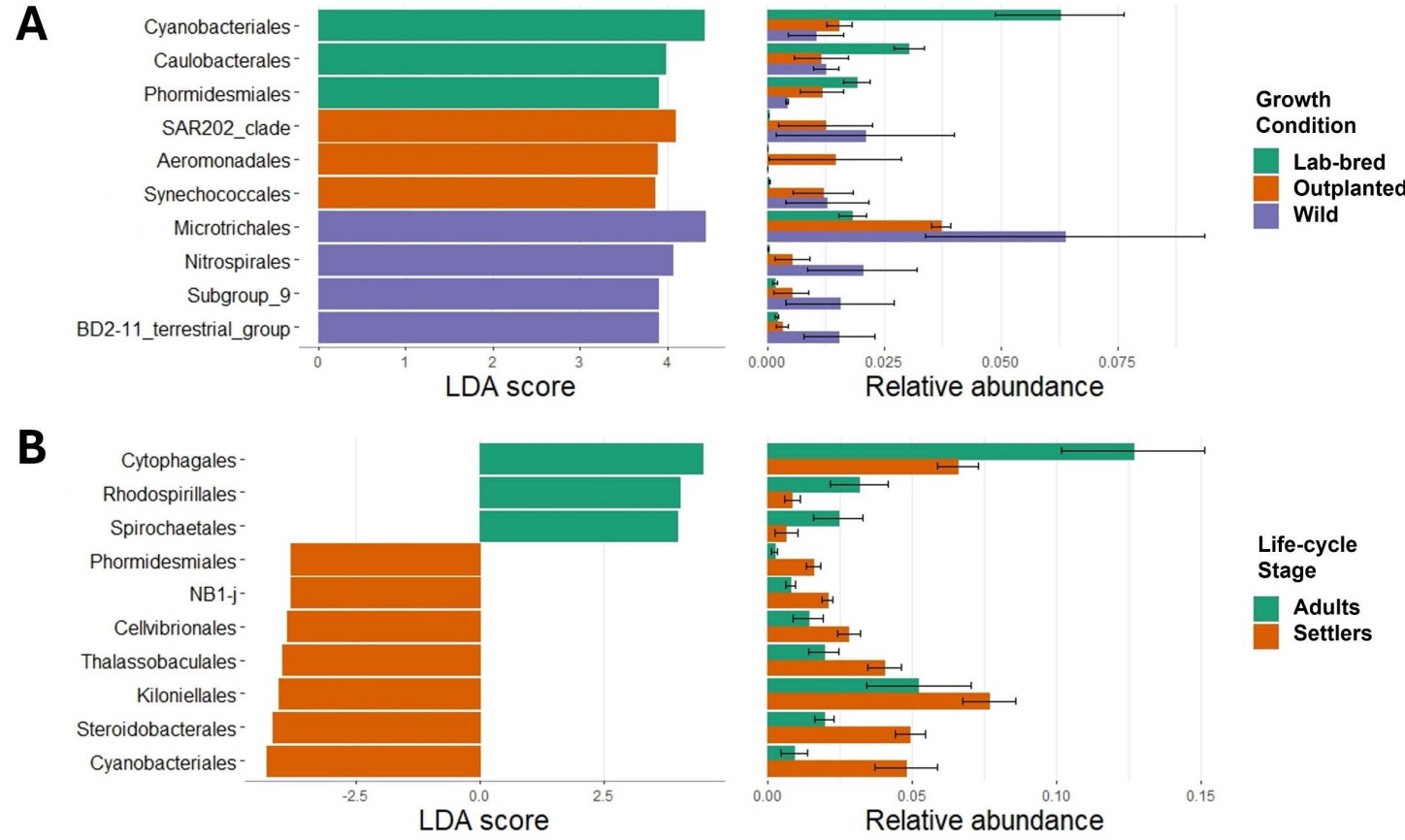

**Fig 3. LEfSe analysis of bacteria associated with _O. faveolata_ at the Order level, displaying taxa with differential abundances (LDA score threshold = 2) and their relative abundances, by settler growth condition (A) and life-cycle stage (B).** A detailed functional prediction suggested by FAPROTAX is shown in Supplementary Table 2 in S1 File. The three most predicted processes for outplanted settlers were aerobic chemoheterotrophy (14.66%), oxygenic photoautotrophy (3.32%), and fermentation (2.33%) followed by nitrogen fixation (0.37%) as the least predicted. For wild settlers, the most predicted processes were aerobic chemoheterotrophy (12.90%), fermentation (2.69%) and oxygenic photoautotrophy (1.79%), with nitrogen fixation (0.20%) as the least predicted.

## Lab-bred settler associated bacteria

The most abundant taxa among the lab-bred _O. faveolata_ settlers were the Orders Kiloniellales, Cytophagales, and Rhodobacterales. The Order Kiloniellales has been previously reported as a chemoheterotrophic taxon [26] and its abundance may explain the most predicted metabolic function for these samples, aerobic chemoheterotrophy. The Order Cytophagales has been described primarily in enriched environments and characterized as a cellulose degrader, in association with the Order Flavobacteriales [27,28], which may potentially impact the microalgal symbionts and disrupt coral health. Such disruption may trigger the release of dimethylsulphoniopropionate (DMSP) by the microalgae promoting the proliferation of the opportunistic Order Rhodobacterales [29], another abundant taxon within these samples. Members of this Order are known for their prevalence in stressed or diseased corals and their ability to detect and degrade DMSP [30]. Although members of this Order have been reported under stressful conditions, several studies have also identified them as part of healthy coral microbiomes [31,32].

The bacterial community composition associated with this group of samples displayed a rare biosphere comprising nearly 50%. These may serve as reservoirs of genetic resources and drive key functions, including nutrient cycling and

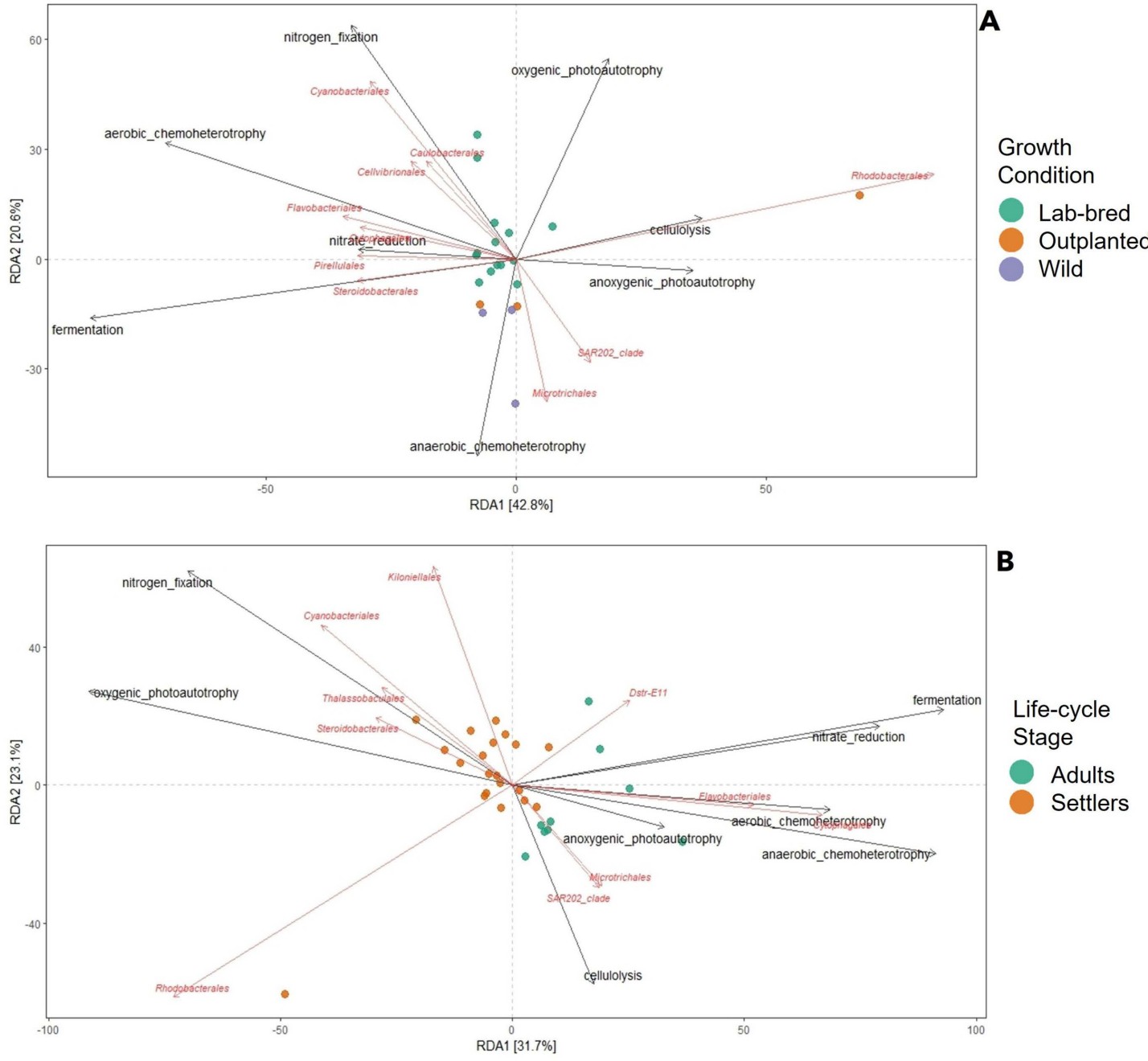

**Fig 4. Redundancy Analysis (RDA) of functional prediction according to FAPROTAX data matrix at bacterial taxa Order level for *O. faveolata*.** Grouped by **(A)** by settler growth condition and **(B)** by coral life-cycle stage.

pollutant degradation, community assembly, and microbiome-host health [33]. However, it remains unclear whether the rare biosphere contains additional taxa for specific functions ensuring that these are not lost in a changing environment, or whether these taxa perform a unique function, despite their low abundance [34,35].

## Comparison by settler growth conditions

The lab-bred *O. faveolata* settlers were characterized by the presence of the order Cyanobacteriales, which has been reported as an early colonizer in various coral species [36,37]. The significance of this order during early coral life stages may be that it enhances the efficiency of essential nutrient cycling, particularly for nitrogen and sulfur [36,38,39]. Another representative taxon within this group was the order Phormidesmiales, known for providing UV protection for the microalgae and as a potential oxygen supply [40]. The order Caulobacterales emerged as a primary contributor to heterotrophy in corals, facilitating the decomposition of organic matter through the breakdown and cycling of organic compounds [41,42]. According to the FAPROTAX results, oxygenic photoautotrophy and nitrogen fixation were the most predicted processes for this group of samples indicating that the presence of these taxa within the LBOFS group might be supporting metabolic processes essential for coral development during the early stages of the life cycle.

For the outplanted *O. faveolata* settlers, the order SAR202_clade emerged as the most representative taxon. It has earlier been reported in the healthy tissue of adults of *O. faveolata* from the same geographic region [43]. However, its ecological role in corals remains poorly understood as it is predominantly found in the microbiome of marine sponges [44–47] suggesting that its presence may primarily result from horizontal transmission. Nevertheless, its presence may contribute benefits such as organic matter degradation and nutrient translocation to the coral tissue because it contains genes related to ammonia reduction and assimilation, as well as sulfur incorporation for amino acid production [48,49] This information could explain why nitrate reduction was the most predicted function in these samples based on the FAPROTAX results (Supplementary Table 2 in S1 File).

The orders Microtrichales and Nitrospirales were the most representative in the wild settler samples. The order Microtrichales has primarily been documented in industrial activated sludge wastewater [50] suggesting that this group may be contributing to nutrient cycling through organic matter degradation via carbon mineralization and complex substrate hydrolysis [51] or through the anammox process under high concentrations of nitrite and ammonium [52]. Elevated nitrite levels in the environment support the growth of the order Nitrospirales [53], another representative taxon within the wild settler group. In terms of function, fermentation and anaerobic chemoheterotrophy were the most predicted processes. These processes may indicate that these taxa facilitate nutrient cycling under low-oxygen conditions.

## Comparison by life-cycle stage

The bacterial composition of the lab-bred *O. faveolata* settlers exhibited similarities at the Class level with previous reports for *O. faveolata* adults [43,54,55]. These findings suggest that *O. faveolata* may harbor a core bacterial community composed mainly by the Classes Alphaproteobacteria, Gammaproteobacteria, and Bacteroidia across life-cycle stages (Supplementary Fig 4 in S1 File).

The orders Cytophagales and Rhodospirillales within the adult samples displayed the highest differential abundances. The order Cytophagales is characterized by its association with nutrient-enriched environments [27] with its potential role as cellulose degraders [28] and its high abundance in bleached or diseased corals [42,56]. Similarly, the order Rhodospirillales has been identified as an opportunistic and potentially pathogenic taxon frequently reported in stressed corals [57]. Given that Scleractinian coral tissue loss disease (SCTLD) severely affected reefs of the Mexican Caribbean in recent years [58,59], the presence of these orders may be related to this event and suggest that this coral species remains susceptible under the prevailing environmental conditions.

This study has some limitations that should be considered. The number of outplanted and wild settlers were limited because it was not easy to locate and collect them in the field. Moreover, the samples were directly collected and preserved in RNAlater without prior rinsing; the absence of this step may have introduced potential contamination or bias in the bacterial composition results, possibly reflecting some microorganisms from the surrounding water column. Therefore, the results should be interpreted with caution. Nevertheless, despite this limitation, this work provides the first exploratory insights into *O. faveolata* lab-bred settlers' microbiome and establishes a baseline for future research.

## Conclusion

A well-represented cyanobacterial community was detected in LBOFS samples. It may contribute to nitrogen fixation, oxygenic photoautotrophy, and efficient cycling of nutrients such as nitrogen and sulfur, which are crucial for development. However, the presence of the orders Microtrichales, Nitrospirales, Cytophagales, and Rhodospirillales in settlers found in the wild and adult samples, along with the high representation of predicted metabolic processes such as anaerobic chemoheterotrophy, fermentation, nitrate reduction, and cellulolysis, may collectively suggest an increased in coral susceptibility to diseases, bleaching and other harmful conditions, which may negatively impact them. These findings highlight that bacterial taxa and predict metabolic processes that may contribute to coral settler functioning. The exploratory nature of this work, provides a valuable baseline for future research, including the isolation and application of beneficial bacterial consortia to enhance settler survival rates, which may be crucial in coral restoration efforts. Furthermore, this study contributes to advancing knowledge of microbial ecology in the Mexican Caribbean.

## Supporting information

**S1 File.**   Supplementary Table 1. PERMANOVA for bacterial community structures associated with *O. faveolata* by (A) settler growth conditions and (B) life-cycle stage. Supplementary Table 2. Functional prediction suggested by FAPROTAX by growth condition. Supplementary Table 3. Kruskal-Wallis by settler growth condition. Supplementary Table 4. Functional prediction suggested by FAPROTAX by life-cycle stage. Supplementary Table 5. Kruskal-Wallis by life-cycle stage. Supplementary Table 6. Read counts obtained for each *O. faveolata* sample after quality filtering. Supplementary Table 7. Number of ASV presented in each *O. faveolata* sample. Supplementary Fig. 1. Rarefaction curves at 4800 reads across *O. faveolata* samples. The curves indicate that sequencing depth was sufficient to capture most of the bacterial diversity within the dataset. (PCF01-PCF15 corresponded to LBOFS; PCF21-PCF32 to adult colonies; PCF34-PCF36 to outplanted settlers and PCF37-PCF39 to wild settlers). Supplementary Fig. 2. Alpha diversity estimated by Shannon index and ANOVA analysis for bacterial communities associated with *O. faveolata.* A) settler growth conditions and B) life-cycle stages. Supplementary Fig. 3 Functional prediction of bacterial communities associated with *O. faveolata* lab-bred settlers, according to the FAPROTAX database. Supplementary Fig. 4 Bacterial community composition at class level across all *O. faveolata* samples. Relative abundances of the 10 major bacterial classes are shown. Rare biosphere is grouped under "Others".
(DOCX)

## Acknowledgments

The authors would like to thank Sandra Mendoza-Quiroz and Eduardo Ávila-Pech for their field assistance during sample collection, and Abril Gamboa-Muñoz for her lab assistance.

## Author contributions

**Conceptualization:** Amanda Pérez-Trejo, Ma. Leopoldina Aguirre-Macedo, Anastazia T. Banaszak, José Q. García-Maldonado.

**Data curation:** Amanda Pérez-Trejo.

**Formal analysis:** Amanda Pérez-Trejo.

**Funding acquisition:** Anastazia T. Banaszak, José Q. García-Maldonado.

**Investigation:** Amanda Pérez-Trejo, Ma. Leopoldina Aguirre-Macedo, Anastazia T. Banaszak, José Q. García-Maldonado.

**Methodology:** Amanda Pérez-Trejo.

**Project administration:** Anastazia T. Banaszak, José Q. García-Maldonado.

**Resources:** Anastazia T. Banaszak, José Q. García-Maldonado.

**Software:** Amanda Pérez-Trejo.

**Supervision:** Ma. Leopoldina Aguirre-Macedo, Anastazia T. Banaszak, José Q. García-Maldonado.

**Validation:** Amanda Pérez-Trejo, Ma. Leopoldina Aguirre-Macedo, Anastazia T. Banaszak, José Q. García-Maldonado.

**Visualization:** Amanda Pérez-Trejo.

**Writing – original draft:** Amanda Pérez-Trejo.

**Writing – review & editing:** Amanda Pérez-Trejo, Ma. Leopoldina Aguirre-Macedo, Anastazia T. Banaszak, José Q. García-Maldonado.

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
