## [Decision Letter · Decision Letter 0]

16 Jul 2025

Dear Dr. García-Maldonado,

Thank you for submitting your manuscript to PLOS ONE. After careful consideration, we feel that it has merit but does not fully meet PLOS ONE’s publication criteria as it currently stands. Therefore, we invite you to submit a revised version of the manuscript that addresses the points raised during the review process.

Please ensure that line numbers are included in the revised manuscript to facilitate the review process.

We look forward to receiving your revised manuscript.

Kind regards,

Parviz Tavakoli-Kolour

Academic Editor

PLOS ONE

2. Please note that your Data Availability Statement is currently missing [the repository name and/or the DOI/accession number of each dataset OR a direct link to access each database]. If your manuscript is accepted for publication, you will be asked to provide these details on a very short timeline. We therefore suggest that you provide this information now, though we will not hold up the peer review process if you are unable.

Additional Editor Comments (if provided):

Reviewers' comments:

Reviewer's Responses to Questions

**Comments to the Author**

1. Is the manuscript technically sound, and do the data support the conclusions?

Reviewer #1: Partly

Reviewer #2: Partly

Reviewer #3: Partly

2. Has the statistical analysis been performed appropriately and rigorously?

Reviewer #1: Yes

Reviewer #2: Yes

Reviewer #3: Yes

3. Have the authors made all data underlying the findings in their manuscript fully available?

Reviewer #1: Yes

Reviewer #2: Yes

Reviewer #3: Yes

4. Is the manuscript presented in an intelligible fashion and written in standard English?

Reviewer #1: No

Reviewer #2: Yes

Reviewer #3: Yes

Reviewer #1: General comments:

This article investigates the bacterial communities associated with the scleractinian coral Orbicella faveolata, with emphasis on settlers bred under laboratory conditions. These communities are compared between outplanted and wild settlers, as well as with adult colonies. The authors employed 16S rRNA gene amplicon sequencing, supported their analyses with relevant statistics, and examined predicted microbial metabolic processes.

The study addresses a relevant and timely topic with potential implications for future coral restoration strategies. The methodology is generally sound, and the analyses are suitable for the questions posed. The findings indicate that outplanted lab-bred Orbicella faveolata settlers harbor distinct bacterial communities compared to their wild counterparts, with metabolic functions related to nitrogen fixation being increased in the former Differences in microbial composition and metabolic potential between settlers and adult colonies are also reported.

However, the manuscript would benefit from significant improvements in its structure, clarity, and interpretation. The current organization and phrasing make it difficult to fully appreciate the novelty of the findings, and in places, the conclusions overreach the data. A more cautious and evidence-based discussion is needed.

Additionally, the manuscript requires minor language corrections related to grammar, punctuation (e.g., missing commas), formatting (e.g., inconsistent spacing), and italics of Latin names following established scientific conventions. Line numbers should be added to facilitate the review process, and supporting information should be moved to the end of the manuscript in accordance with the journal’s submission guidelines.

Major concerns:

Abstract. The authors begin by stating lab-bred Orbicella faveolata settlers exhibit low survival rates after outplanting. However, this claim is not supported by any cited studies, either in the introduction or discussion. The authors should provide references to support this statement or modify the text to respect discussed literature.

Introduction. The introduction provides a brief summary addressing the appropriate general topics for the research presented; however, it lacks clarity and suffers from redundancy in how the information is presented. I recommend restructuring the introduction to improve logical flow and coherence while minimizing repetition. A clearer structure could involve introducing the ecological relevance of coral settlement and the known vulnerability of early life stages, and highlighting the impact of coral-microbiome associations, with particular focus on the role of bacteria during early development. Only after establishing this context, present existing literature relating to the impact of bacteria of larval settlement/survival and the rationale for studying the settler microbiome. This could then be followed up with the identification of key knowledge gaps and the study’s objectives.

To further strengthen both the introduction and discussion, the authors should consider incorporating recent studies on external influences affecting Orbicella faveolata larval settlement (e.g.,https://doi.org/10.1371/journal.pone.0292474; https://doi.org/10.1098/rspb.2023.1476;
https://doi.org/10.3354/meps14739). Incorporating these studies alongside the currently presented literature on the microbiome of Orbicella faveolata, and, where available, the microbiome and functional roles associated with coral settlers, would help to highlight the specific knowledge gap their study addresses. This would contextualize the significance of their findings within the broader context of Orbicella faveolata settlement research.

Methods. At present, the methods section lacks sufficient detail to allow full reproducibility. Several key aspects require clarification or elaboration.

A brief explanation of the breeding conditions for the lab-bred settlers should be included in the manuscript, especially considering that the primary reference cited for these procedures is not in English.

Sampling details including collection depth, GPS coordinates, water temperature, sampling time of the year, and the procedures followed between underwater sampling and storage in RNAlater solution. These environmental and procedural factors could influence microbial communities and are important for enabling comparisons with future studies.

It is unclear whether the samples rinsed before storage to reduce seawater-associated microbial contamination. If this was done, it should be described explicitly. If not, the authors should address how potential contamination was handled, especially considering that no control samples were reported.

Sample identifiers should be provided to improve traceability and link sequencing data to samples in results.

The classification of lab-bred Orbicella faveolata settlers and whether it includes or excludes outplanted settlers is unclear. These groups are presented separately throughout the manuscript, which seems appropriate, but the criteria used for this categorization should be explicitly stated.

Bioinformatic analysis requires restructuring and greater detail. Please provide references for QIIME2, DADA2, and Silva v138 database. Specify the DADA2 version used and describe the parameters used throughout the pipeline and downstream processing (e.g., trimming parameters). If default settings were used, this should be clearly stated. Indicate the rarefaction depth applied to the data and justify this threshold, as it affects diversity estimates and comparisons.

The verification that the data meets the assumptions of the ANOVA test (normality and equal variance) should be clarified in the methods section for scientific rigor.

Please clarify how many permutations were used in the PERMANOVA, and if any correcting method was applied.

Results. The methods section states that 34 samples were collected, yet only 29 were sequenced. This discrepancy should be clarified. Additionally, in the sentence: “A total of 29 samples were sequenced for the 16S rRNA gene, corresponding 14 to LBOFS, three outplanted settlers, three wild settlers, and three wild adults”, only 23 samples are accounted for. The authors should clarify the source and classification of all 29 samples.

It is unclear whether the term “characterization of LBOFS-associated bacteria” includes the outplanted settlers. The inconsistency in sample numbers throughout the manuscript: 16 mentioned in Methods, 14 in Results, 15 in Fig 1, 13 in S1 Fig 1, and 14 in S2 Fig 1A, needs to be resolved and clearly explained.

Figure 1 and S1 Fig1 – The legends should more detailed information to guide the reader. Please specify the coral species studied and clarify whether the 10 major bacterial orders were determined across all samples or per sample. Also state the data type used (e.g., ASV table based on rarefied relative abundances), and explain how the “rare biosphere” was defined. If the rare biosphere refers to orders not within the top 10, this definition may be misleading. Many studies define the rare biosphere using relative abundance thresholds (e.g., 0.1% or 0.01% per sample), and there are specialized tools for this purpose (e.g., https://doi.org/10.1038/s42003-025-07912-4). The authors are encouraged to clarify or reconsider their terminology here.

The analysis would benefit from also presenting results at the family level, which is common in coral microbiome studies. This would provide depth and complexity to the study and is feasible since the data are available from the same ASV table already in use.

Figure 2 – The legend should provide further details to guide the reader. Clearly indicate whether the data were rarefied (and if so, at what read depth), whether any transformation (e.g., Hellinger) was applied, and include the PERMANOVA F and p values. For consistency, “Type” should be renamed to “Life-cycle stage.”

PERMANOVA analysis by settlers’growth conditions should be followed up with pairwise comparisons to identify which groups differ significantly. If only lab-bred and wild settlers were compared, this should be explicitly stated. These results are critical to support the manuscript’s conclusions.

Figure 3 – The figure caption should inform the reader on the coral species studied, which statistical tests were used, if any normalization method was applied, nd whether the error bars represent standard deviation or standard error.

Please ensure that supplementary figures and tables are consistently numbered in the order they appear in the manuscript. Currently, the manuscript jumps from S4_Table1 to S7_Table1. Furthermore, S6_Fig1 is only mentioned in the discussion section, but appears essential to the central analysis; it should be referenced earlier in the Results and may be better suited as a main figure or combined with Fig 1 to show a more complete taxonomic comparison across groups.

Figure 4 – The caption should be more detailed, please see previous suggestions. Caption should also explain that the ten most abundant orders are also represented, and what values were used for the length of assossiated arrows in the RDA plot.

Discussion + Conclusions. The discussion regarding Rhodobacterales appears somewhat speculative and may overstate the implications of the presented results. While this bacterial order is frequently associated with dysbiotic states, it is also recognized as a ubiquitous component of the marine environment and has been reported in multiple studies as a part of healthy coral microbiomes (e.g., https://doi.org/10.1093/femsec/fix143,
https://doi.org/10.3389/fmicb.2016.00458). If there were no signs of disease or stress, its presence, especially given its detection in both wild settlers and adult colonies, may reflect a typical, non-pathogenic component of the coral holobiont. Linking predicted functions with the presence of specific taxa should be approached with caution, particularly when inferred at order level. I suggest rephrasing the discussion.

The sentence “However, it raises questions about how these conditions might impact the microalgae symbionts, suggesting that further studies are needed.” feels out of place, as the role of microalgal symbionts is not a focus of this study and their involvement is not supported by the presented data. Unless the authors provide specific data or rationale linking microbial shifts to changes in symbiont health, this statement should be removed or revised to avoid unnecessary speculation.

Clear conclusions regarding the aims of the study raised in the introduction are missing. While the study highlights microbial differences across life stages and growth conditions, it does not assess coral health, growth, or survival outcomes, which are critical for evaluating the feasibility of lab-bred settlers in restoration efforts. Therefore, the implications of these findings for coral reef restoration appear premature based on the current results. Regarding the statement “This knowledge is a valuable baseline for future research to enhance their survival rates through the isolation and application of beneficial bacterial consortium.”, the data do not support conclusions about functional benefits or improvements in settler health, nor do they provide evidence supporting the use of specific bacterial taxa in applied microbial consortia. If the authors intend to suggest that lab-bred settlers may be viable for restoration, this needs to be stated more clearly, and the first sentence of the abstract should be rephrased. However, without comparative survival data or experimental testing of microbial effects, such an argument remains speculative. I recommend rephrasing the conclusion to reflect the exploratory nature of this work and to suggest directions for future functional or experimental studies that could build on these findings.

The manuscript does not address the limitations of the study

Data availability. While raw sequencing data is shared, it is standard practice to also provide at least the Amplicon Sequence Variant table after quality control, sample metadata, and a summary table indicating the number of reads retained per sample after DADA2 processing and filtering. This enhances transparency and utility of the dataset for future research.

Minor concerns:

Abstract. The abbreviation “PCoA” should be removed from the abstract, as it is not used again in that section.

In the sentence “Processes such as nitrogen fixation and oxygenic photoautotrophy were significantly different for settlers compared to adults, and nitrate reduction and anaerobic chemoheterotrophy for adults compared to settlers.”, the phrasing would be more accurate and informative if the authors clarified whether these processes were enriched in one group or the other. “Significantly different” is vague without specifying the direction of the change.

Introduction. The phrase “has been proposed” should be supported by appropriate references.

The manuscript should use consistent terminology. For example, the authors should standardize the use of “larval settlement” instead of “larvae settlement”, and of “16S rRNA gene” instead of “16s rRNA” throughout the manuscript.

Methods. Please specify which Qubit fluorometer model and DNA quantification kit (e.g., Broad-Range or High-Sensitivity) were used.

Bray-Curtis is not a measure of distance, please replace this with “dissimilarity measures”.

Mentioning of ANOVA analysis applied to the Shannon index is missing in the results section although mentioned in the discussion.

The manuscript does not indicate whether archaeal ASVs were removed from the dataset. Given that the primer pair used may amplify archaeal sequences, the authors should clarify whether archaeal reads were present and filtered out. If not, the study should more accurately address prokaryotic community instead of bacterial community.

Results. Percentage symbols (“%”) are missing. Additionally, please specify whether the reported percentages represent average values.

When reporting Kruskal-Wallis test results for predicted functions, it would be more accurate and informative to describe the functions as enriched in settlers or adults, rather than simply stating they were significantly different.

Reviewer #2: This manuscript examines the microbial composition of early life stages of lab-bred corals compared to in situ corals of different ages. While the topic area is valuable, the manuscript lacks the depth needed to evaluate the results. The study also suffers from relatively low sample numbers.

Major concerns:

METHODS

Substantially more details are needed throughout the methods. For example, what is a "settler"? What age and size are settlers? Where did the settlers come from? The abstract indicates that they are "lab-bred" - provide the details about how this was achieved. Provide a brief description of the culture conditions, currently there is just a reference given to a paper in Spanish describing culture conditions for Acropora palmata. Provide details of how samples were collected in the field. Version numbers are needed for all software packages and databases. For example, Silva 138 is not sufficient detail, as there are substantial differences between v. 138.1 and 138.2.

RESULTS

The manuscript states that 34 samples were collected, but only 29 were sequenced (and the NCBI Bioproject has 46 biosamples). Please explain why not all samples were successfully sequenced.

Provide an assessment of sequencing quality: what was the total sequencing depth? how many reads per sample?

Define how the rare biosphere was determined. Why is this concept important to this study?

DISCUSSION

The discussion should include comparisions to other papers describing the Orbicella faveolata microbiome as well as papers examining the microbiomes of early life stages of coral.

Minor comments:

RESULTS

All figures in the reviewer pdf are very low resolution.

REFERENCES

Overall, the references are poorly formatted and many references are missing details like volume and page numbers.

Reviewer #3: In this manuscript, García-Maldonado and colleagues compared microbiome variation in lab-bred, outplanted and wild coral settlers, as well as between settlers and adult invidividuals, and further performed functional prediction of bacterial communities. The topic is interesting and novel. However, the manuscript could benefit from some additions to the results section, and generally more precise language when drawing conclusions in the discussion.

I also have a general comment: I am not sure if this is a journal requirement, but the manuscript does not have line numbers and that makes the review process a bit difficult – I strongly suggest adding line numbers in the revised version. Below are other suggestions.

Abstract

“By the life-cycle stage, the Order Cytophagales was the most representative of adult samples, and Cyanobacteriales for settlers.“ – does “representative“ mean ‚more or less abundant in‘?

Introduction

Last paragraph question 3) – please specify if you mean adults and settlers in the lab or wild.

Results

“Characterization of LBOFS communities“ – ASV richness should be reported and analysed in addition to Shannon index, as this allows an assessment of both richness and evenness of the communities. It can be informative especially to compare across life stages and settler conditions.

Discussion

Last paragraph of “Lab-bred settler associated bacteria“ section: “These results suggest that the composition of the rare biosphere within these samples may be crucial for settler functioning, suggesting that further analysis of this bacterial fraction is warranted, though beyond the scope of this study.“ – This conclusion is not obvious from the data. Is it based on a comparison of the proportion of rare taxa between lab-bred and other settlers? Please explain the specific result and how this conclusion is drawn from it.

“The bacterial composition of the lab-bred Orbicella faveolata settlers exhibited similarities with previous reports for O. faveolata adults [36,47,48]. These findings suggest that O. faveolata may harbor a core bacterial community across life-cycle stages (Data in S6_Fig1).“ – Please give a specific explanation of how your results are similar to those of other reports in corals – which specific taxa, which patterns (adults versus settlers) etc.

Figures

Fig. 1 could have more informative labels at the bottom of the bar – maybe some subheadings to indicate which groups the bars belong to.

Fig. 4 Font size should be enlarged – it‘s impossible to see any of the functions in the figure.

**Do you want your identity to be public for this peer review?** For information about this choice, including consent withdrawal, please see our Privacy Policy

Reviewer #1: No

Reviewer #2: No

Reviewer #3: No

---

## [Author Response · Author response to Decision Letter 1]

1 Oct 2025

September, 2025

Emily Chenette

Editor-in-Chief Plos One

Dear Dr. Chenette:

We appreciate the opportunity for a resubmission of our research article entitled “Differentiation of the bacterial communities associated with Orbicella faveolata across different growth conditions and life-cycle stages”. We appreciate your comments to improve this work. Below, we provide a point-by-point response to all the received comments.

Reviewer #1 General comments:

This article investigates the bacterial communities associated with the scleractinian coral Orbicella faveolata, with emphasis on settlers bred under laboratory conditions. These communities are compared between outplanted and wild settlers, as well as with adult colonies. The authors employed 16S rRNA gene amplicon sequencing, supported their analyses with relevant statistics, and examined predicted microbial metabolic processes.

The study addresses a relevant and timely topic with potential implications for future coral restoration strategies. The methodology is generally sound, and the analyses are suitable for the questions posed. The findings indicate that outplanted lab-bred Orbicella faveolata settlers harbor distinct bacterial communities compared to their wild counterparts, with metabolic functions related to nitrogen fixation being increased in the former Differences in microbial composition and metabolic potential between settlers and adult colonies are also reported.

However, the manuscript would benefit from significant improvements in its structure, clarity, and interpretation. The current organization and phrasing make it difficult to fully appreciate the novelty of the findings, and in places, the conclusions overreach the data. A more cautious and evidence-based discussion is needed.

Additionally, the manuscript requires minor language corrections related to grammar, punctuation (e.g., missing commas), formatting (e.g., inconsistent spacing), and italics of Latin names following established scientific conventions. Line numbers should be added to facilitate the review process, and supporting information should be moved to the end of the manuscript in accordance with the journal’s submission guidelines.

Answer: We sincerely appreciate Reviewer 1’s constructive feedback and thoughtful suggestions. We made improvements in the structure, clarity and interpretation of the manuscript. Also, different organization of the information was presented to highlight the novelty of the findings. Moreover, improvements in the discussion were made based on the suggestions. Finally, we have applied the suggested grammatical and formatting corrections, and standardized the terminology, as well as corrected italics and capitalization throughout the manuscript.

Reviewer #1 Major concerns:

Reviewer #1 Abstract. The authors begin by stating lab-bred Orbicella faveolata settlers exhibit low survival rates after outplanting. However, this claim is not supported by any cited studies, either in the introduction or discussion. The authors should provide references to support this statement or modify the text to respect discussed literature.

Answer: This statement was modified in the abstract (L15-18), to supporting the statement; post-settlement survival studies have been incorporated in the introduction (L76-89)

Reviewer #1 Introduction. The introduction provides a brief summary addressing the appropriate general topics for the research presented; however, it lacks clarity and suffers from redundancy in how the information is presented. I recommend restructuring the introduction to improve logical flow and coherence while minimizing repetition. A clearer structure could involve introducing the ecological relevance of coral settlement and the known vulnerability of early life stages, and highlighting the impact of coral-microbiome associations, with particular focus on the role of bacteria during early development. Only after establishing this context, present existing literature relating to the impact of bacteria of larval settlement/survival and the rationale for studying the settler microbiome. This could then be followed up with the identification of key knowledge gaps and the study’s objectives.

Answer: The introduction has been restructured according to the following order previously suggested (L48-99): “Ecological relevance of coral settlement and the known vulnerability of early life stages, and highlighting the impact of coral-microbiome associations, with particular focus on the role of bacteria during early development. Only after establishing this context, present existing literature relating to the impact of bacteria of larval settlement/survival and the rationale for studying the settler microbiome”

“INTRODUCTION

Coral-microbiome associations across all life stages are essential for survival, particularly in a changing environment. Larvae settlement is fundamental to coral restoration efforts using assisted larval propagation techniques. However, larval post-settlement survival is a major bottleneck due to the susceptibility of settlers to predation, competition, and stochastic disturbances. Improving larval growth rates could help them pass through this vulnerable phase (1,2). The analysis of the microbiome could be crucial for enhancing coral function and used to improve efforts to rescue threatened species (3). In addition, the response of the microbiome to stressful conditions and environmental changes enables the identification of key host-microbiome associations, which may be essential for coral resilience and adaptation (4–6). Coral microbiome monitoring and manipulation can be used to effectively manage and maintain organismal health(3).

Bacteria are the dominant group in the coral microbiome as they contribute to stress tolerance and adaptation to a changing environment (7). This occurs through mechanisms including the promotion of coral nutrition and growth, mitigation of stress and toxic compound impacts, deterrence of pathogens, and benefiting early life-stage development (8). During the early-life stages, beneficial traits include the production of signals for the modulation or regulation of larval settlement, the production of biofilm, which facilitates larval settlement, and quorum quenching or the breakdown of quorum-sensing molecules, which disrupt cell-cell signaling of beneficial microbes for other species competing for space (8).

As previously reviewed by (9) practices associated with coral breeding for reef restoration focus on improving settler health prior to outplanting on the reef, including the use of beneficial microbes. This approach requires studying the microbiome during the early life stages of corals to potentially improve larval post-settlement survival and growth using probiotics.

Previous studies on O. faveolata have shown that microbial communities associated with crustose coralline algae (CCA) enhance larval settlement (10,11). In reef environments, degrading conditions can reduce CCA cover and may compromise this process. Other studies conducted in different coral species at early life stages reported the promotion of larval settlement and enhancement of post-settlement survival using probiotics. For example, when administering the strain cB07 of Metabacillus indicus to Pocillopora damicornis (12). Higher post-settlement survival was exhibited by the coral Porites astreoides when settled onto bacteria biofilm as well as on coralline algae (13). In another study, larval settlement/metamorphosis of Pocillopora damicornis was strongly correlated with the bacterial community composition of diverse calcified algae (14). Bacterial strains isolated from the crustose coralline algae Hydrolithon reinboldii were tested on Leptastrea purpurea larvae and found to enhance settlement behavior (15).

Based on the above, there is limited information on bacterial communities associated with settlers bred under culture conditions and their ability to improve settler health once outplanted onto coral reefs. Therefore, as a first step, this study aimed to: (1) identify and analyze the microbial communities associated with Orbicella faveolata lab-bred settlers (LBOFS), (2) compare the bacterial communities associated with LBOFS grown under different conditions, and (3) compare life-cycle stages (settlers vs wild adults). With this information, we aimed to identify key bacterial taxa and metabolic processes to be considered for further research and testing of microbial consortia to improve post-settlement survival rates and growth once outplanted onto the reef.”

Reviewer #1 Introduction. To further strengthen both the introduction and discussion, the authors should consider incorporating recent studies on external influences affecting Orbicella faveolata larval settlement

(e.g.,https://doi.org/10.1371/journal.pone.0292474; https://doi.org/10.1098/rspb.2023.1476;
https://doi.org/10.3354/meps14739).

Answer: The suggested references were revised and cited to strengthen the introduction. (L76-79)

“Previous studies on O. faveolata have shown that microbial communities associated with crustose coralline algae (CCA) enhance larval settlement (10,11). In reef environments, degrading conditions can reduce CCA cover and may compromise this process.”

Reviewer #1 Methods. At present, the methods section lacks sufficient detail to allow full reproducibility. Several key aspects require clarification or elaboration.

A brief explanation of the breeding conditions for the lab-bred settlers should be included in the manuscript, especially considering that the primary reference cited for these procedures is not in English.

Answer: A new section was incorporated into the Methods to provide a comprehensive description of the breeding conditions. “Sample collection and breeding conditions” (L 102-120)

“Sample collection and breeding conditions

During the spawning of Orbicella faveolata, gametes were collected and carefully transported to the CORALIUM Laboratory at the Unidad Academica de Sistemas Arrecifales in Puerto Morelos, Quintana Roo, Mexico. Upon arrival, assisted fertilization was applied to promote the fusion of gametes from distinct colonies, to generate new individuals.

Embryonic development was monitored by microscopy post-fertilization. Embryos were transferred to incubators and maintained in the laboratory until larval development, which occurred 20-32 hours after fertilization. Incubators were maintained at 28°C with a 12:12 hours light:dark photoperiod.

Larval settlement occurred between 3 and 7 days post-fertilization. During this stage conditioned substrates were introduced into the incubators to promote settlement. These substrates were composed of a mixture 50:50 of sea sand and concrete, that were previously conditioned on the reef for 60 days, to allow for the attachment of microorganisms relevant to larval settlement.

After settlement, they were transferred to outdoor aquarium facilities where light intensity was kept below 400 µmol m⁻² s⁻¹ during the first two months, with gradual increases thereafter. Salinity was maintained at 35-36 PSU throughout this period. All this information can be found in (16).”

Reviewer #1 Methods. Sampling details including collection depth, GPS coordinates, water temperature, sampling time of the year, and the procedures followed between underwater sampling and storage in RNAlater solution. These environmental and procedural factors could influence microbial communities and are important for enabling comparisons with future studies.

Answer: The sampling details, including depth, GPS coordinates, year, and settlers' dimensions, were incorporated to improve clarity (L 122-132)

“From a total of 34 samples of Orbicella faveolata, 16 samples corresponded to lab-bred settlers (LBOFS) (PCF01-PCF16). The remaining samples were collected from Jardines Reef (20° 49’ 51.53” N 86° 52’ 29.92” W) at a depth of 9 m in the Puerto Morelos Reef National Park, Mexico in 2022. For adult colonies, 12 fragments (< 1 cm2) were collected (PCF21-PCF32). For outplanted settlers, 3 samples from the 2019 cohort were collected (PCF734-PCF736), along with 3 additional samples from wild settlers (< 1 cm2) (PCF737-PCF739). The settlers were carefully detached from the substrate using a sterile scalpel and placed into Eppendorf tubes with RNAlater solution (Qiagen, Hilden, Germany) to preserve genetic material, as were the samples of the adult colonies.”

Reviewer #1 Methods. It is unclear whether the samples rinsed before storage to reduce seawater-associated microbial contamination. If this was done, it should be described explicitly. If not, the authors should address how potential contamination was handled, especially considering that no control samples were reported.

Answer: The samples were directly preserved in RNAlater, without undergoing a rinsing process. This limitation has been included in the study limitations paragraph (L 381-389)

“This study has some limitations that should be considered. The number of outplanted and wild settlers were limited because it was not easy to locate and collect them in the field. Moreover, the samples were directly collected and preserved in RNAlater without prior rinsing; the absence of this step may have introduced potential contamination or bias in the bacterial composition results, possibly reflecting some microorganisms from the surrounding water column. Therefore, the results should be interpreted with caution. Nevertheless, despite this limitation, this work provides the first exploratory insights into O. faveolata lab-bred settlers' microbiome and establishes a baseline for future research.”

Reviewer #1 Methods. Sample identifiers should be provided to improve traceability and link sequencing data to samples in results.

Answer: Samples identifiers were added in Lines 122-129. Sequencing data is declared in the section “Data Availability Statement” under the BioProject accession number PRJNA1244432.

“From a total of 34 samples of Orbicella faveolata, 16 samples corresponded to lab-bred settlers (LBOFS) (PCF01-PCF16). The remaining samples were collected from Jardines Reef (20° 49’ 51.53” N 86° 52’ 29.92” W) at a depth of 9 m in the Puerto Morelos Reef National Park, Mexico in 2022. For adult colonies, 12 fragments (< 1 cm2) were collected (PCF21-PCF32). For outplanted settlers, 3 samples from the 2019 cohort were collected (PCF734-PCF736), along with 3 additional samples from wild settlers (< 1 cm2) (PCF737-PCF739).”

Reviewer #1 Methods. Bioinformatic analysis requires restructuring and greater detail. Please provide references for QIIME2, DADA2, and Silva v138 database. Specify the DADA2 version used and describe the parameters used throughout the pipeline and downstream processing (e.g., trimming parameters). If default settings were used, this should be clearly stated.

Answer: We have added bioinformatic details, including tool references, versions and trimming parameters, which are now described in the “Bioinformatic analysis section” (L 167-173)

“Demultiplexed paired-end 16S rRNA gene sequences were analyzed using the Quantitative Insights into Microbial Ecology pipeline (QIIME2 version QIIME 2 2024.10.1) (18). The DADA2 (V 1.16) tool was employed to filter and assess the sequencing reading quality with trimming parameters 30 to 200 (19), while the SILVA 138.1 database was used for taxonomy assignment, resulting in Amplicon Sequence Variant (ASV) (20). Sequences assigned to the domain Archaea were removed from the analysis.”

Reviewer #1 Methods. Indicate the rarefaction depth applied to the data and justify this threshold, as it affects diversity estimates and comparisons.

Answer: The rarefaction parameter was added (L 177).

“The dataset was rarefied to a depth of 4800 prior to the calculation of the abundance tables.”

Reviewer #1 Methods. Please clarify how many permutations were used in the PERMANOVA, and if any correcting method was applied.

Answer: The number of permutations was added in L 183.

“Principal Coordinates Analysis (PCoA) based on Bray-Curtis dissimilarity along with permutational multivariate analysis of variance (PERMANOVA) with 999 permutations, were used to assess differences in the bacterial community composition.”

Reviewer #1 Results. The methods section states

---

## [Editor Report · Decision Letter 1]

13 Oct 2025

Differentiation of the bacterial communities associated with Orbicella faveolata across different growth conditions and life-cycle stages

PONE-D-25-19351R1

Dear Dr. García-Maldonado,

We’re pleased to inform you that your manuscript has been judged scientifically suitable for publication and will be formally accepted for publication once it meets all outstanding technical requirements.

Kind regards,

Parviz Tavakoli-Kolour

Academic Editor

PLOS ONE
---

## [Editor Report · Acceptance letter]

PONE-D-25-19351R1

PLOS ONE

Dear Dr. García-Maldonado,

I'm pleased to inform you that your manuscript has been deemed suitable for publication in PLOS ONE. Congratulations! Your manuscript is now being handed over to our production team.

Kind regards,

on behalf of

Dr. Parviz Tavakoli-Kolour

Academic Editor

PLOS ONE